# Preparation of Nanofiber Bundles via Electrospinning Immiscible Polymer Blend for Oil/Water Separation and Air Filtration

**DOI:** 10.3390/polym14214722

**Published:** 2022-11-04

**Authors:** Yin Tang, Tang Zhu, Zekai Huang, Zheng Tang, Lukun Feng, Hao Zhang, Dongdong Li, Yankun Xie, Caizhen Zhu

**Affiliations:** 1Institute of Low-Dimensional Materials Genome Initiative, College of Chemistry and Environmental Engineering, Shenzhen University, Shenzhen 518060, China; 2College of Textile Science and Engineering (International Institute of Silk), Zhejiang Sci-Tech University, Hangzhou 310018, China

**Keywords:** electrospinning, nanofiber bundles, polymer blend, oil/water separation, air filtration

## Abstract

Nanofiber bundles with specific areas bring a new opportunity for selective adsorption and oil/water or air separation. In this work, nanofiber bundles were prepared by the electrospinning of immiscible polystyrene (PS)/N-trifluoroacetylated polyamide 6 (PA6-TFAA) blends via the introduction of carbon nanotubes (CNTs) or a copolymer of styrene and 3-isopropenyl-α, α’-dimethylbenzene isocyanate (TMI), which was denoted as PS-co-TMI. Herein, CNT was used to increase the conductivity of the precursor for enhancing the stretch of PS droplets under the same electric field, and PS-co-TMI was used as a reactive compatibilizer to improve the compatibility of a PS/PA6-TFAA blend system for promoting the deformation. Those obtained nanofiber bundle membranes showed an increase in tensile strength and high hydrophobicity with a water contact angle of about 145.0 ± 0.5°. Owing to the special structure, the membranes also possessed a high oil adsorption capacity of 31.0 to 61.3 g/g for different oils. Moreover, it exhibits a high potential for gravity-driven oil/water separation. For example, those membranes had above 99% separation efficiency for silicon oil/water and paraffin wax/water. Furthermore, the air filtration efficiency of nanofiber bundle membranes could reach above 96%, which might be two to six times higher than the filtration efficiency of neat PS membranes.

## 1. Introduction

Nanofiber membranes have high specific surface areas, highly interconnected pore structures, nano scale pore sizes, the potential to incorporate active chemistry on a nanoscale, and low initial solidification [1]. However, the fibers in the nonwoven mats are randomly arranged, which results in low mechanical strength and other disadvantage. In order to improve the desirable properties of nanofibers and expand the application of nanofiber membranes, more and more research tend to the fabrication and application of aligned nanofiber bundles or yarns. The nanofiber bundles and yarns are of significant value in a wide range of applications, including tissue engineering [2,3], drug release [4,5], sensors [6,7,8], reinforced composites [9,10] and filtration [3,11].

Electrospinning is one of most simple and versatile ways to fabricate nonwoven membranes with fibers varying from micro to nano scales. With different devices and methods, nanofiber bundles could be achieved by electrospinning [12,13,14], including the auxiliary electrode collecting method [15,16,17], self-bundling electrospinning method [4,18], water bath-collecting method [19,20], etc. For example, Wang and his co-workers [13] produced self-assembling nanofiber yarn bundles via a swirling collector. When the electrospinning started, the nanofibers broke off from the jet and whipped from the collector, as the collector kept swirling. The nanofiber bundles formed between the fiber tips and the ground plate by whipping around the anchor point and contact adjacent bundles. Guan et al. [21] prepared PA66 fiber bundles with two opposite electrode pins as the collector. After post-treatment with 0.05 wt% multiwall carbon nanotubes (MWCNTs), the electric conductivity and tensile strength rapidly increased to 0.2 S/cm and 103 MPa. Wang et al. [22] fabricated polyacrylonitrile (PAN) fiber yarns with the self-bundling electrospinning method. Compared with normal nonwoven PAN membranes, the tensile strength, tensile module and elongation at break of the fiber yarn membrane increased more than 320%, 2670% and 260%, respectively.

Recently, functional fibers with complex structures, such as islands-in-the-sea [23], cocontinuous [24] and core–sheath [25] structures could be gained by the electrospinning of polymer blends through controlling the phase separation. For example, Tang et al. [26] have prepared PS/PA6 bead-free fibers by electrospinning of the PS/PA6-TFAA blend, which was ascribed to the good electrospinnability of PA6-TFAA at low concentration. Furthermore, the core–shell superfine fibers could be prepared by introducing with an interfacial compatibilizer, which could promote the dispersion of minor components during microphase separation. Wang et al. [27] electrospun liquid crystal/polymer core–sheath fibers formed by phase separation. They could change the width of the polymer sheath, and the diameter of the liquid crystal core increases with controlling the feed rates. In addition, they found that the viscosity gradient resulted in an inward movement of the lower viscosity component, which lead to the core/sheath structure. However, there has been little effort made to prepare nanofiber bundles toward the control of phase separation in electrospinning.

The membrane technique, especially electrospun membranes, has been widely used in the separation of oil–water mixtures with property of high oil removal efficiency and stable effluent quality, which have been applied in food processing, pharmaceutical desalination and fuel cell industries [1,28]. A high hydrophobicity and specific surface could improve the selective adsorption and separation efficiency in water-in-oil mixtures. Nonwoven fabric also is the main product for filtration in air pollution with low cost, light weight and high filtration efficiency [29,30]. The size and structure of fibers and specific surface of membranes are the key to the filtration efficiency [31,32,33].

In this study, we used immiscible polystyrene (PS)/N-trifluoroacetylated polyamide 6 (PA6-TFAA) blending with different additives as a model system to fabricate nanofiber bundles after post-treatment with formic acid etching. With a proper amount of conductive particles or compatibilizers as additives, the force of PS droplets in fluid jets had been enhanced, and droplets could be largely stretched, which resulted in nanofiber bundles. Due to the more aligned fiber structure of nanofiber bundles, the higher tensile strength and tensile module were observed. Moreover, the hydrophobicity and selective adsorption for oil had been improved, resulting from the rougher surface morphology and higher specific area. These nanofiber bundle membranes showed significant application in water-in-oil separation and air filtration.

## 2. Materials and Methods

### 2.1. Material

Polystyrene (PS, Mw, 228,800 g/mol) and polyamide (PA6, Mw, 49,400 g/mol) were purchased from Yangzi-BASF Styrenics Company (Hong Kong, China) and UBE Nylon Ltd. (Taphong, Thailand), respectively. Styrene (St), benzoyl peroxide (BPO), toluene, methanol, dichloromethane (CH_2_Cl_2_), chloroform (CHCl_3_), Karl Fischer reagent (without pyridine), silicon oil and paraffin wax were purchased from Sinopharm Chemical Reagent Co., Ltd. (Shanghai, China) Sunflower oil was purchased from Luhua Group Co., Ltd. (Shanghai China) 3-Isopropenyl-α,α’-dimethylbenzene isocyanate (TMI) and trifluoroacetic anhydride (TFAA) were purchased from Aladdin Co., Ltd. (Fukuoka, Japan) Carbon nanotubes (CNT) were purchased from Chengdu Organic Chemicals Co. Ltd. (Chengdu, China) Methylene blue and oil red O were purchased from Shanghai Macklin Biochemical Co., Ltd. (Shanghai, China) PA6 was used after dried at 80 °C in vacuum, and all other chemicals were used as received.

### 2.2. Preparation of PS-Co-TMI

A random copolymer of St and TMI, denoted as PS-co-TMI, was synthesized by the free radical copolymerization of St and TMI in toluene with BPO as a free radical initiator. The products were precipitated twice in methanol and then filtered and dried under the vacuum. In this work, the Mn of PS-co-TMI was 37,600 g/mol. Details of the synthesis of PS-co-TMI can be found in other papers [34,35,36].

### 2.3. N-Trifluoroacetylation of PA6

A given amount of PA6, CH_2_Cl_2_ and TFAA were added to a flask. The molar ratio between TFAA and the amide group of PA6 was about 1.5:1. The volume ratio of CH_2_Cl_2_ and TFAA was 2:1. After reacting for 12 h at 25 °C, the PA6-TFAA was gained by a rotary evaporator.

### 2.4. Fabrication of Fiber Membranes

The precursor of PS/PA6-TFAA was prepared by dissolving PS and PA6-TFAA in CHCl_3_ with a concentration of 0.2 g/mL, respectively, and then blended with a volume ratio of 1:1. Five different solutions were prepared varying CNT or PS-co-TMI with 0, 2 and 4 wt%. The electrospinning solution was placed in a 5 mL syringe attached to a stainless needle with an internal diameter of 0.6 mm. The needle was connected to the positive voltage of 12 KV, and the rotary collector was connected to negative voltage of -3 KV, which was applied by a high-voltage power supply (DW-P303-1ACD8, Tianjin Dongwen High-voltages Source Co., Tianjin, China). The distance between the needle tip and collector was maintained at 15 cm, and the flow rate was 3 mL/h, which was controlled by a syringe pump (KDS-200, KD Scientific Inc., Holliston, MA, USA). The spun membranes were dried at 80 °C in vacuum for 6 h to remove residual solvents along with the deacetylation of PA6-TFAA. Then, PS/PA6-TFAA fibers turned out to be PS/PA6 fibers. After that, the membranes were etched by formic acid three times and each time took 24 h, which were marked as e-PS/PA6, e-PS/PA6/2PS-co-TMI, and e-PS/PA6/2CNT with 2 wt% of additives, and e-PS/PA6/4PS-co-TMI, e-PS/PA6/4CNT with 4 wt% of additives.

### 2.5. Oil Selective Adsorption Measurement

The oil adsorption tests were carried on at 25 °C. The spun membranes were placed in a Petri dish containing 30 mL water and 1 g oil. The oil was dyed red by oil red O. After 1 hour’s adsorption, the samples were drained for 2 min and then weighed. The adsorption capacity of membranes was evaluated in three kinds of oil: silicon oil, sunflower oil and paraffin wax. The oil adsorption capacity of membranes was determined from the following equation:(1)$$Q=\frac{m_{s}-m_{0}}{m_{0}}$$
where Q was the oil adsorption capacity (g/g), *m_s_* was the total mass of wet adsorbent (g), and *m*_0_ was the weight of initial adsorbent (g).

### 2.6. Oil/Water Separation Performance

The spun membrane was fixed in between two both-sides-opening tubes with an inner diameter of 3.3 cm. Polyester film was put under the filter as a supporting membrane. The oil/water mixtures were placed in a glass beaker with a mass ratio of 1:100, which was followed by vigorously stirring for 1 h. The mixtures were poured into the upper tube, and the separation carried on due to gravity. Three types of oil/water mixtures, such as silicon/water, sunflower oil/water and paraffin wax/water were analyzed. The water was dyed blue by methylene blue in the silicon/water and paraffin wax/water mixtures.

The separation efficiency of the spun membranes was calculated by the formula:(2)$$A=\frac{M_{0}-M_{1}}{M_{0}}\times 100\%$$
where A was separation efficiency (%), *M*_1_ was water content levels after separation determined by Karl Fischer titration (g/g), and *M*_0_ was water content levels before separation by Karl Fischer titration (g/g).

### 2.7. Air Filtration Performance

The spun membrane was fixed in between two one-side-opening tubes with an inner diameter of 3.2 cm. Polyester film was put under the filter as a supporting membrane. One of the tubes was connected with the smoke generator and the other one was connected with the smoke detector. The smoke generator could offer smoke with a concentration of 15–25 mg/m^3^ and wide size distribution from <100 nm to 2 μm. The smoke flow rate used in the efficiency test was 5 L/min.

The air filtration efficiency of the spun membranes was calculated by the formula:(3)$$\eta =\frac{C_{0}-C_{1}}{C_{0}}\times 100\%$$
where *η* was the separation efficiency (%), *C*_1_ was the smoke concentration after separation (mg/m^3^), and *M*_0_ was the smoke concentration before separation (mg/m^3^).

### 2.8. Characterization

The AC resistance of the precursors was tested by an electrochemical workstation (CHI660E, Shanghai Chenhua Instrument Company, Shanghai, China) with the three-probe method, using a graphite electrode/calomel electrode/graphite electrode. The sinusoidal AC signal was set as 0.5 mV, the frequency range was set as 10^−2^~10^6^ Hz, and the testing temperature was set as 25 °C. Conductivity was calculated by the formula:(4)$$\sigma =\frac{d}{R\times S}$$
where σ was the conductivity of electrospinning solution (S/cm), *d* was the distance of the graphite electrode (cm), *R* was the self-resistance of the precursor (Ω), and *S* was the area of the graphite electrode below the liquid level (cm^2^).

The morphologies of electrospun fiber membranes were observed by using scanning electron microscopy (SEM, ZEISS ULTRA 55, Carl Zeiss AG, Oberkochen, Germany) with an accelerating voltage of 5 KV. Before the test, the samples were dried for 2 h in a vacuum oven at 80 °C and then gold sputtered for 4 min. The mechanical properties of membranes were tested using an electronic universal testing machine (UTM2102, Shenzhen Suns Technology Stock Co., Ltd., Shenzhen, China) at a deformation rate of 10 mm/min. The length and width of the samples were 20 and 5 mm. The water contact angle (WCA) with a water volume of 3 μL was measured by a contact angle measuring device (OCA 20, DataPhysics Instruments, Filderstadt, Germany). The water content levels were determined by Karl Fisher titration. The smoke concentration was detected by ELPI (Dekati Ltd., Kangasala, Finland). The pressure drop could be calculated from the height difference marked in the tube U.

## 3. Results

### 3.1. Morphologies of Electrospun Fibers

To understand the structure of PS/PA6 fibers, the SEM images of as-spun fibers and fibers etched by formic acid are displayed in Figure 1. Smooth PS/PA6 fibers with average diameters of 2.0 μm were obtained, as shown in Figure 1a. After being etched by formic acid, the PS/PA6 fibers became PS fibers and some grooves turned out on the surface, as shown in Figure 1b. However, nanofibers with a diameter of about 200 nm only displayed on the core of e-PS/PA6 fibers, which was observed on the cross-section of the etched fibers in Figure 1c.

In the emulsion-like immiscible precursors, PA6-TFAA formed the continuous phase, while PS formed the discontinuous phase [37,38,39], resulting from the different viscosity of PA6-TFAA and PS. The viscosities of PS and PA6-TFAA were 141 and 20 mPa·s. The polymer with lower viscosity preferred to form the continuous phase, which meant PA6-TFAA formed the continuous phase. Furthermore, PS and PA6-TFAA performed phase separation during the electrospinning process due to the incompatibility [23,40]. While in electrospinning, the electric force applying on PA6-TFAA was larger than that on PS, and the velocity gradient formed perpendicular on the interface of PS and PA6-TFAA domains due to different electrospinnability. In the core of fluid jets, high velocity led to PA6-TFAA offering a larger stretch for PS droplets, and the PS droplets stretched, head-to-tail coalesced, and formed the nanofibers. Nevertheless, on the shell of fluid jets, PS droplets did not gain enough stretch before coalescence and formed the solid shells, as the velocity on the shell was lower than that on the core.

In order to fabricate the perfect nanofiber bundles, two kinds of additives were blended with PS/PA6-TFAA for enhancing the stretch of PS droplets, respectively, which were CNT and PS-co-TMI. Figure 2 showed the conductivity of PS/PA6-TFAA solution with an increment of CNT. With higher conductivity, the fluid jets could gain a larger stretch under the same electric field. In the fluid jets, the random PS droplets were stretched, turned from sphere to ellipsoid and became close head-to-tail. Next, PA6-TFAA between the head and tail of ellipsoid-like PS droplets was squeezed out, and the PS droplets coalesced as rod-like droplets. With coalescence going, PS nanofibers were formed before the solvent disappeared. The surface morphology of PS/PA6 fibers almost remained smooth after introducing CNT, as shown in Figure 3a,d. After being etched by formic acid, the PS nanofiber bundles appeared and were oriented along the fiber on the surface, as shown in Figure 3b,c,e,f. The diameter of nanofibers of bundles was 50–150 nm. As the conductivity of PS/PA6-TFAA with 2 wt% CNT was too low to offer enough stretch to PS droplets in fluid jets, rod-like PS domains were observed (Figure 3b,c).

On the other hand, the isocyanate groups of PS-co-TMI could react with the terminal group of PA6-TFAA to form a graft copolymer of PS-*g*-PA6-TFAA, which acted as a compatibilizer in the PS/PA6-TFAA system and enhanced the interfacial interaction. From Figure 4a, when the amount of PS-co-TMI was 2 wt%, parts of the surface appeared porous, because some PS migrated to the surface, and a high evaporated solution was used in the precursor [41,42]. After being etched by formic acid, the porous surface remained, the smooth surface transformed to coherent nanofibers, and the cores were filled with nanofibers, which were imperfect nanofiber bundles (Figure 4b,c). When the content of PS-co-TMI increased to 4 wt%, the interfacial interaction between PS and PA6-TFAA had improved. As PA6-TFAA had higher spinnability, more electric force was applied on the PA6-TFAA domain under the same electric field. Therefore, more drag force was transferred from the PA6-TFAA domain to PS droplets when the amount of compatibilizer increased. The PS droplets gained more stretch and transformed from sphere to ellipsoid, rod and nanofibers finally.

Figure 5 displays the tensile stress vs. strain curves for PS and e-PS/PA6 without and with additives. For all samples, the tensile stress–strain curves showed a non-linear elastic behavior until the highest tensile stress. That is because the fibers slipped and oriented under stress loading. When the stress reached its maximum, a drastic reduction in tensile stress happened in e-PS/PA6 with additives. That means the orientation of bundles had completed in the stress loading direction, and the bundle membranes broke at the point of the highest stress. However, the tensile stress decreased slowly after the highest stress for neat PS and e-PS/PA6 fiber membranes, resulting from incomplete orientation of the random fibers. Under the same electric field, PS prefers to form the random fibers because of unstable fluid jets action due to lower electrospinnability. By becoming introduced with 4 wt% CNT, the conductivity of electrospun solution increased almost twice (Figure 2), which enhanced the electrospinnability of PS and formed orientated fibers. On the other hand, the electrospinnability of PA6-TFAA is high enough, resulting in forming more orientated fiber membranes. By adding 4 wt% PS-co-TMI, the compatibilizer could improve the interfacial interaction between PS and the PA6-TFAA domain, which could efficiently enhance the stability of fluid jets and formed orientated fiber membranes.

The tensile strength and tensile module of different fiber membranes are summarized in Figure 6. The tensile strength and tensile module of PS membranes are lower than 0.10 MPa and 4.9 MPa, respectively. After adding PA6-TFAA into the precursor, the orientation of fibers in the membranes had greatly improved, and the structure changed from bead-on-string to “imperfect” nanofiber bundles, by which the tensile strength increased to 0.89 MPa and the tensile module increased to 20.3 MPa. When blended with 4 wt% PS-co-TMI or CNT into PS/PA6-TFAA, the orientation of fibers had further improved, and perfect nanofiber bundles were formed, which made the tensile strength and tensile module 53–58 times and 13–17.5 times higher than that of neat PS fiber membranes.

### 3.2. Air Filtration Performance of Electrospun Fiber Membranes

The filtration performance of membranes was investigated by the smoke of moxibustion tobacco. As shown in Figure 7a, the filtration efficiency of PS and e-PS/PA6 increased from 14 to 54% and 25 to 72%, respectively. However, the filtration efficiency of these membranes was much lower than that of e-PS/PA6/4PS-co-TMI and e-PS/PA6/4CNT, which varied from 88 to 96% and 74 to 89% with the increment of basis weight. Meanwhile, the pressure drop of PS membranes increased from 20 to 60 Pa, which were lowest in these four kinds of membranes (Figure 7b). That means the packing density of PS membranes was low, and this fluffy structure facilitated air flow penetrating through the membranes, resulting in low filtration efficiency. The pressure drop of e-PS/PA6 and e-PS/PA6/4CNT was almost the same, which was between 65 and 130 Pa. The higher filtration efficiency of e-PS/PA6/4CNT could be ascribed to the nanofiber bundle structure. The pressure drop of e-PS/PA6/4PS-co-TMI greatly rose from 150 to 320 Pa with a basis weight of 2.6 to 3.4 g/m^2^; then, it slightly increased from 320 to 430 Pa with a basis weight of 3.8 to 7.7 g/m^2^, which means the packing density of e-PS/PA6/4PS-co-TMI was the highest in these four kinds of membranes. That helps to further improve the filtration efficiency.

In order to figure out the air filtration behavior, four kinds of membranes with a basis weight of about 5.4 g/m^2^ were chosen for the next discussion, which were denoted as XXX-5, for example PS-5 membrane. The details of membranes are shown in Table 1. The packing density (α) [43] and the quality factor (QF) [44] are described by the following equation:(5)$$\alpha =\frac{W}{\rho \times Z}$$
(6)$$\mathrm{QF}=-\frac{\ln\left(1-\eta \right)}{\Delta P}$$
where *W* is the basis weight of membranes, *ρ* is the density of the polymer material, *Z* is the thickness of membranes, *η* is the air filtration efficiency, and Δ*P* is the pressure drop.

The smoke of moxibustion tobacco has a wide size distribution from <100 nm to 2 μm. As Figure 8a shows, the majority of particles was between 0.8 and 2 μm, whose concentration exceeded 3 mg/m^3^. The thickness of the PS-5 membrane was the largest (47.6 μm, as Table 1 shown), which could hardly stop smoke particles with diameter <300 nm penetrating, and the highest filtration efficiency was just about 30%, which might be attributed to the low packing density [45]. The gap of filtration efficiency between e-PS/PA6/4CNT-5 and e-PS/PA6-5 had broadened when the diameter of smoke particles was lower than 2 μm. That means the surface structure of nanofiber bundles could capture more particles, especially small particles with almost the same packing density and pressure drop (as Table 1 shows). The filtration efficiency of e-PS/PA6/4PS-co-TMI-5 was above 90% for diameter >480 nm and about 85% for diameter <320 nm, as it possessed the highest packing density and special structure of nanofiber bundles. However, the QF of e-PS/PA6/4PS-co-TMI-5 was the lowest because of the highest pressure drop, as Table 1 shows. Due to the strong balance of filtration efficiency and pressure drop, e-PS/PA6/4CNT-5 seemed to be the best filtration membranes with QF of 1.71 × 10^−4^ Pa^−1^.

Figure 9 shows SEM images of these four kinds of membranes after filtration with 1, 5, 20, and 30 min. After 1 min filtration, the smoke particles were hardly found in PS-5 and e-PS/PA6-5 membranes. Comparing Figure 9c with Figure 9d, most of e-PS/PA6/4PS-co-TMI-5′s bundles had scattered, which increased the specific surface area and improved the packing density of the membranes. This is the reason more particles were noticed in e-PS/PA6/4PS-co-TMI-5 membranes. In addition to spheroidal particles, wizened particles were observed on the nanofiber bundles (inset picture of Figure 9d,g,h), which indicated liquid particles could be partly adsorbed into the bundles and made the bundles remain at a small size. Moreover, the wizened particles could act as branches on the bundles to increase the surface area, which further improved the filtration efficiency. When the filtration took 5 min, the particles captured by fibers become more, as shown in Figure 9e–h. The wizened particles could be observed frequently on the nanofiber bundles and the bundles were partly covered by the smoke dust, as shown in Figure 9g, h. Furthermore, the bundles stuck together by dust and “dust membranes” formed at the intersection of fibers (Figure 9h). At 20 min, the particles on the fibers become larger in membranes of e-PS-5 and e- PS/PA6-5 (Figure 9i,j). The “dust membranes” started to show at the intersection of bundles in e-PS/PA6/4CNT-5 (Figure 9k) and covered almost the whole membranes of e-PS/PA6/4PS-co-TMI-5 (Figure 9l). Finally, the fibers were covered by the spheroidal particles and the size became obviously larger in PS-5 membranes (Figure 9m) because the fiber could not adsorb liquid inside. The smooth surface with a few spheroidal particles and the dust membranes were observed in e-PS/PA6-5 membranes. The membranes with higher filtration efficiency were blocked by dust, as shown in Figure 9o,p.

According to the above results, we supposed electrospun fibers captured the dust when continuous flow brought the dust and hit the fiber membranes. Some dust attached to the fibers. With more smoke feeding, the dust particles were able to move along the fibers [31]. Then, the particles aggregated or captured the new one in air flow to form larger particles. As the filtration continued, the particles became large enough to contact each other and aggregated to form dust membranes at the intersection of two fibers. As time went by, the triangular dust membranes formed, and finally, the whole filters were blocked by dust. Because the particles-captured-ability was too weak, e-PS-5 could build the dust membranes (Figure 9m) within 30 min filtration. In addition, e-PS/PA6-5 membranes were able to build the triangular dust membranes (Figure 9o). Due to the powerful particles-captured-ability, the nanofibers bundle membranes were blocked at 30 min for e-PS/PA6/4CNT-5 and 20 min for e-PS/PA6/4PS-co-TMI-5.

### 3.3. Oil/Water Separation Performance of Electrospun Fiber Membranes

Wettability depends on the chemical composition and surface morphology of the membranes. All the samples shown in Figure 10 were PS membranes, including as-spun neat PS fibers and etched blended electrospun fibers. So, the surface morphology made a difference in the water contact angle (WCA) in this work. The porous, bead-on-string and fluffy structure of as-spun neat PS fibers led to the high WCA of 133.4 ± 1.3°. After adding PA6-TFAA, the WCA of the etched fibers just had a small increase to 139.3 ± 1.3°, resulting from the almost smooth and bead-free fiber structure. With the addition of CNT or PS-co-TMI, the WCA of nanofiber bundle membranes increased to 145.0 ± 0.5°. Although the surface structure of nanofiber bundles was rough in this work, the bead-free and dense surface structure could not make the more abundant hierarchical structure for membranes, which could not further improve the hydrophobicity of membranes.

Figure 11 displays the oil selective adsorption capacities of PS fiber membranes with different morphologies. The neat PS fiber membranes showed the minimum adsorption from 10.1 to 17.1 g/g in different oil (as Figure 11a shown). Although neat PS fiber membranes had a fluffy structure which indicated large external pores, the membranes easily shrunk, and the oil desorbed due to the poor tensile module. The nanofiber bundle membranes exhibited excellent oil adsorption from 31.0 to 61.3 g/g. That can be explained by the rough surface and internal penetrated gap of nanofiber bundles, which led to a bigger specific area that was more easily saturated by oil. In addition, the capillary action was enhanced by the internal gap of the bundles; thus, the oil could be quickly adsorbed into the inner void. As Figure 11b shows, the dyed paraffin wax was adsorbed by the e-PS/PA6/4PS-co-TMI membrane in a few seconds. On the other hand, although there was a large void in the core of etched PS/PA6 without an additive, some of them could not be filled with highly viscous oil, as there were a few penetrated grooves on the solid surface, as shown in Figure 1b,c. That is the reason the oil adsorption of e-PS/PA6 without any additive was almost half of that of the etched PS/PA6 with CNT or PS-co-TMI.

The water-in-oil emulsion separation efficiency is shown in Figure 12. The neat PS fiber membranes have a minimum separation efficiency of 91.8%, 90.0% and 92.2% in silicon oil/water, sunflower oil/water and paraffin wax/water separation, respectively. The separation efficiency of e-PS/PA6 fiber membranes increase a little bit. Due to the highest hydrophobicity and oil adsorption capacity, the separation efficiency of nanofiber bundle membranes increased to above 99.0% in silicon oil/water and paraffin wax/water separation and 97.7% in sunflower oil/water. In this water-in-oil emulsion separation, when oil/water mixture was poured into the upper tube, the oil was immediately adsorbed by the nanofiber bundle membranes and formed an oil film on the surface of the filters, while water was repelled by hydrophobic membranes or the oil film. As a result of gravity force, oil ran through the membrane and collected. In this case, the pictures of the dyed water removed by e-PS/PA6/4PS-co-TMI membranes are shown in Figure 13. After filtration, the oils were clear.

## 4. Conclusions

The nanofiber bundles were prepared by electrospinning PS/PA6-TFAA blended with CNT or PS-co-TMI and then etched by formic acid. CNT could increase the conductivity of the precursor in order to improve the drag force of the electric field for PS droplets in jets. Meanwhile, PS-co-TMI used as a reactive compatibilizer could enhance the interfacial interaction of PS and PA6-TFAA, resulting in drag force efficiently transferred from the PA6-TFAA domain to PS droplets. Due to the greater orientation of fibers, the nanofiber bundle membranes had a high tensile strength and tensile module of 1.7 MPa and 63.0–84.7 MPa. As a result of the greater roughness on the surface of the nanofiber bundle membrane, WCA reached 145.0 ± 0.5°, approaching superhydrophobicity. The oil selective adsorption was from 31.0 to 61.3 g/g, resulting from a high internal void and specific area. In addition, the membranes showed a high water-in-oil emulsion separation of above 99% in silicon oil/water and paraffin wax/water. Moreover, the structure of nanofiber bundles also helps to improve air filtration efficiency. The highest air filtration efficiency of the nanofiber bundle membranes could reach above 96%.

## Figures and Tables

**Figure 1 polymers-14-04722-f001:**
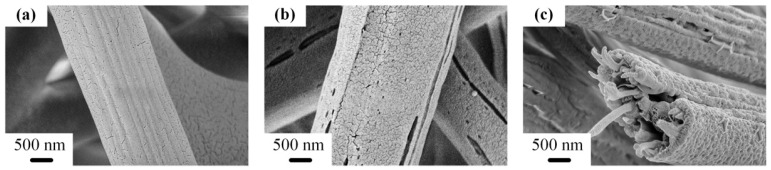
SEM images of PS/PA6 fibers without (**a**) and with (**b**,**c**) etched by formic acid.

**Figure 2 polymers-14-04722-f002:**
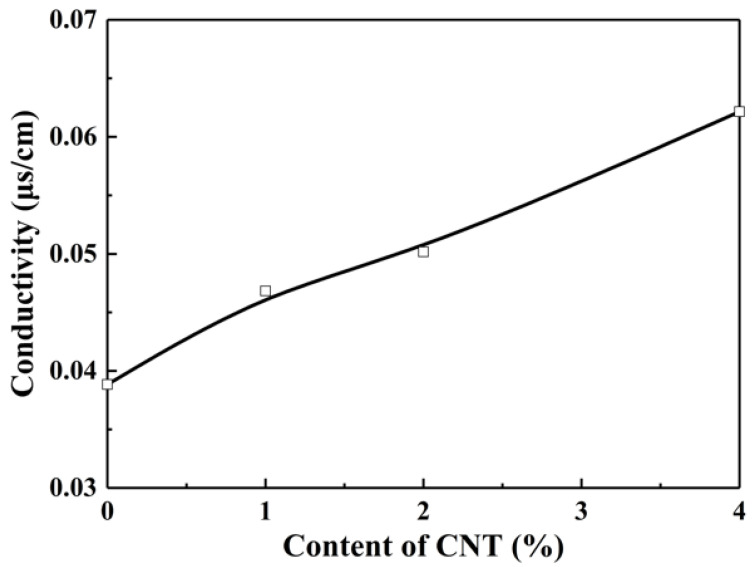
Conductivity of PS/PA6-TFAA electrospinning solution with different content of CNT.

**Figure 3 polymers-14-04722-f003:**
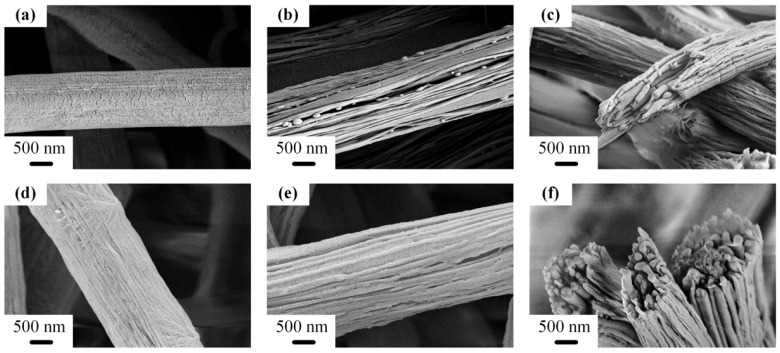
SEM images of electrospun PS/PA6/CNT without (**a**,**d**) and with (**b**,**c**,**e**,**f**) etching by formic acid. The contents of CNT were 2 wt% (**a**–**c**) and 4 wt% (**d**–**f**).

**Figure 4 polymers-14-04722-f004:**
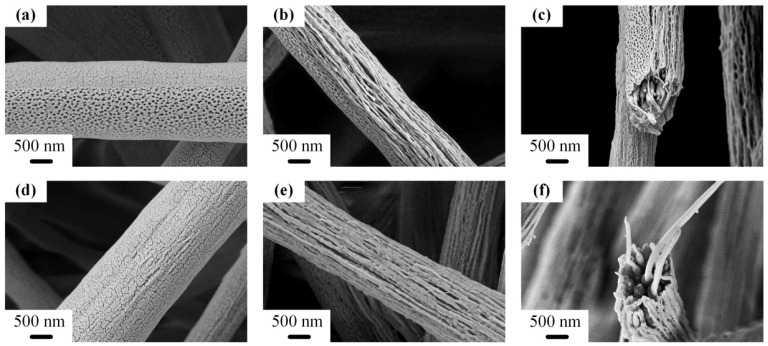
SEM images of electrospun PS/PA6/PS-co-TMI without (**a**,**d**) and with (**b**,**c**,**e**,**f**) etching by formic acid. The contents of PS-co-TMI were 2 wt% (**a**–**c**) and 4 wt% (**d**–**f**).

**Figure 5 polymers-14-04722-f005:**
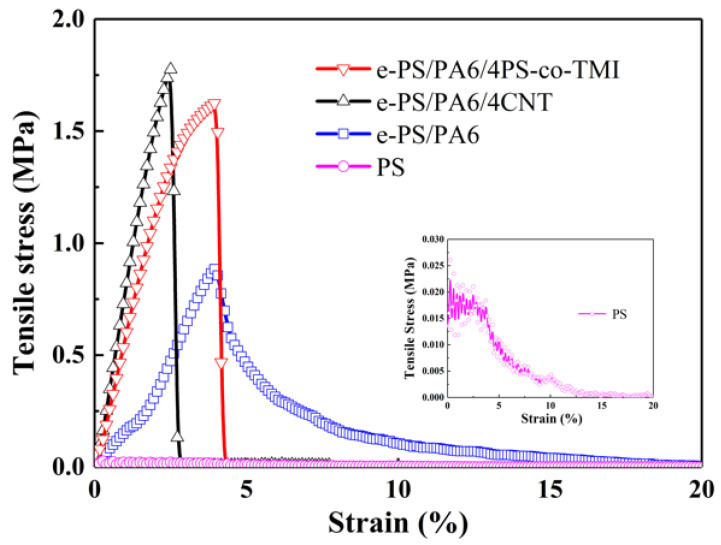
Tensile stress–strain curves of electrospun fibers: PS, e-PS/PA6, e-PS/PA6/4CNT and e-PS/PA6/4PS-co-TMI. The content of CNT or PS-co-TMI was 4 wt%. The fiber membranes electrospun from blended polymer solution were etched by formic acid before being tested.

**Figure 6 polymers-14-04722-f006:**
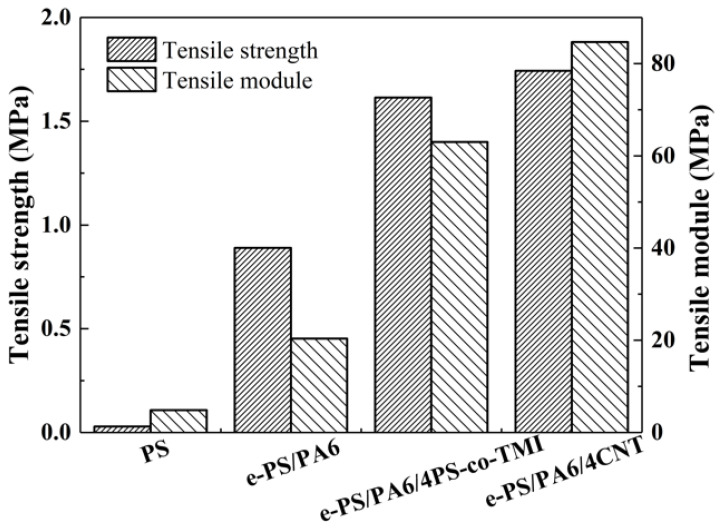
Variation in tensile strength and tensile module of electrospun fibers: PS, e-PS/PA6, e-PS/PA6/4CNT and e-PS/PA6/4PS-co-TMI. The content of CNT or PS-co-TMI was 4 wt%. The fiber membranes electrospun from blended polymer solution were etched by formic acid before tested.

**Figure 7 polymers-14-04722-f007:**
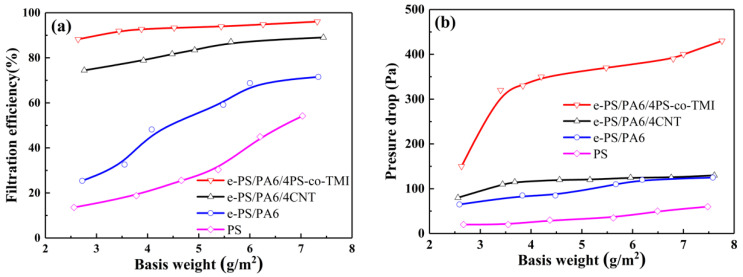
(**a**) Filtration performance of membranes with various basis weight. (**b**) Pressure drops of membranes with various basis weight. The content of CNT or PS-co-TMI was 4 wt%. The fibers electrospun from blended polymer solution were etched by formic acid before being tested.

**Figure 8 polymers-14-04722-f008:**
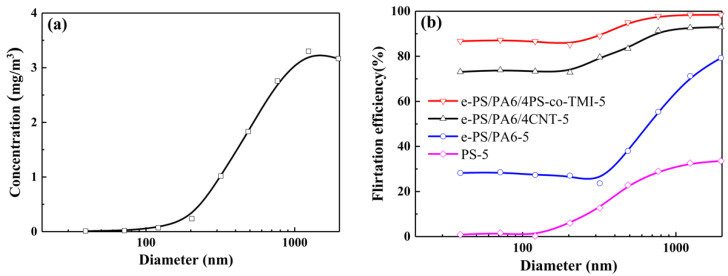
(**a**) Concentration of the smoke particles with different diameter size before filtration. (**b**) Filtration efficiency to different diameter size of smoke particles. The content of CNT or PS-co-TMI was 4 wt%.

**Figure 9 polymers-14-04722-f009:**
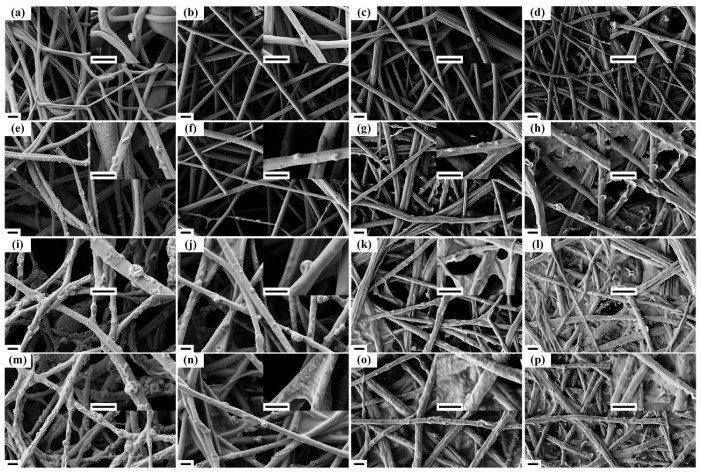
EM images of PS-5 (**a**,**e**,**i**,**m**), e-PS/PA6-5 (**b**,**f**,**j**,**n**), e-PS/PA6/CNT-5 (**c**,**g**,**k**,**o**) and e-PS/PA6/PS-co-TMI-5 (**d**,**h**,**l**,**p**) membranes after filtration. The filtration time was 1 min (**a**–**d**), 5 min (**e**–**h**), 20 min (**i**–**l**) and 30 min (**m**–**p**). The content of CNT or PS-co-TMI was 4 wt%. The scale bar was 5 μm.

**Figure 10 polymers-14-04722-f010:**
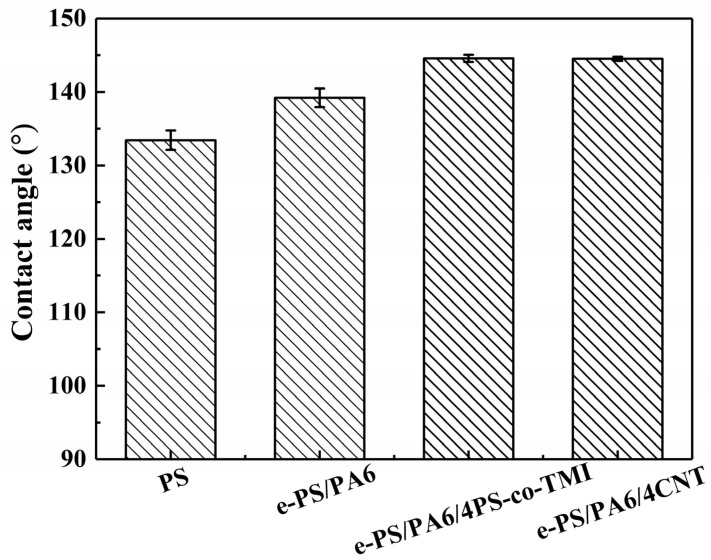
Water contact angle of electrospun fibers: PS, e-PS/PA6, e-PS/PA6/4CNT and e-PS/PA6/4PS-co-TMI. The content of CNT or PS-co-TMI was 4 wt%. The fiber membranes electrospun from blended polymer solution were etched by formic acid before being tested.

**Figure 11 polymers-14-04722-f011:**
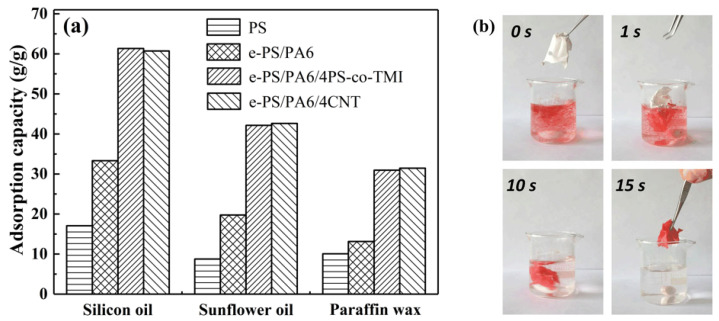
(**a**) The oil selective adsorption of electrospun fibers: PS, e-PS/PA6, e-PS/PA6/4CNT and e-PS/PA6/4PS-co-TMI. (**b**) Optical images of paraffin wax (dyed red with oil red O) selective adsorption by e-PS/PA6/4PS-co-TMI with different time. The content of CNT or PS-co-TMI was 4 wt%. The fiber membranes electrospun from blended polymer solution were etched by formic acid before being tested.

**Figure 12 polymers-14-04722-f012:**
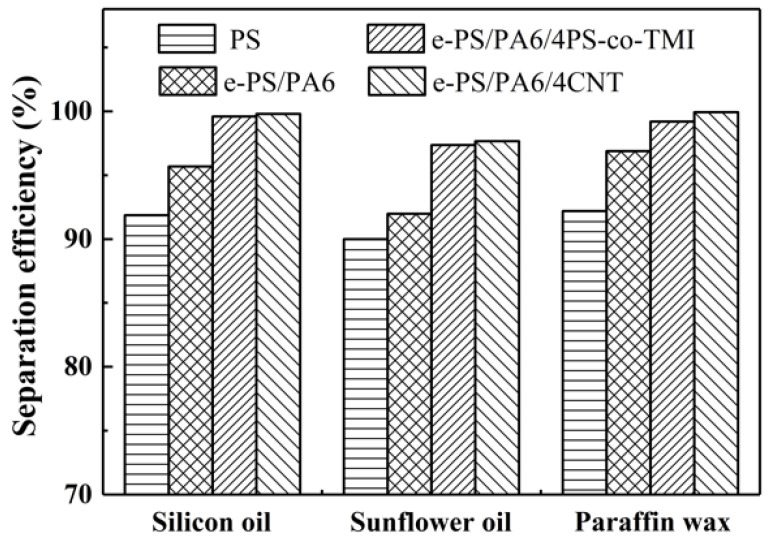
The oil/water separation efficiency of electrospun fibers: PS, e-PS/PA6, e-PS/PA6/4CNT and e-PS/PA6/4PS-co-TMI. The content of CNT or PS-co-TMI was 4 wt%. The fiber membranes electrospun from blended polymer solution were etched by formic acid before being tested.

**Figure 13 polymers-14-04722-f013:**
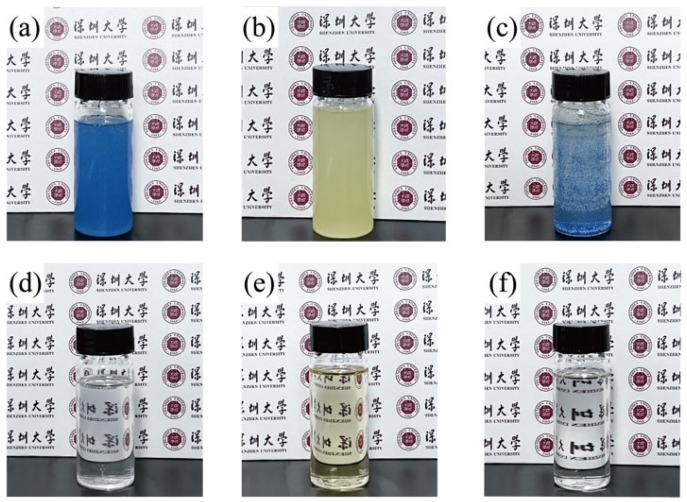
Optical images of water-in-oil emulsion before (**a**–**c**) and after (**d**–**e**) filtration with nanofiber bundle membranes (e-PS/PA6/4PS-co-TMI). Oil/water mixtures were silicon oil/water (**a**,**d**), sunflower oil/water (**b**,**e**) and paraffin wax/water (**c**,**f**). Water was dyed blue with methylene blue in silicon oil/water and paraffin wax/water.

**Table 1 polymers-14-04722-t001:** Filtration performance of membranes.

Samples	Basis Weight (g/m^2^)	Thickness (μm)	Filtration Efficiency (%)	Pressure Drops (Pa)	Packing Density	Quality Factor (×10^−4^∙Pa^−1^)
e-PS/PA6/4PS-co-TMI-5	5.4	28.7	93.9	360	0.18	0.78
e-PS/PA6/4CNT-5	5.6	35.9	87.1	120	0.15	1.71
e-PS/PA6-5	5.5	37.8	59.1	110	0.14	0.81
PS-5	5.4	47.6	30.4	35	0.11	1.03

## Data Availability

No new data were created or analyzed in this study. Data sharing is not applicable to this article.

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
