# Peer review of "Preparation of Nanofiber Bundles via Electrospinning Immiscible Polymer Blend for Oil/Water Separation and Air Filtration"

_polymers, 2022, doi:10.3390/polym14214722_

Round 1

Reviewer 1 Report

In this manuscript, the authors prepared the nanofiber bundle membranes with phrase separation, which exhibited multi-nanofiber-bundle structure and application in Oil/water Separation and Air Filtration. The result is interesting and I think it can be published after a minor revision.

1.      Please unify unit expression, for example, g·g-1 and g/g.

2.      In the oil/water separation experiment, what is the liquid flux?

3.      In the oil absorption experiment (Figure 11), why the same kinds of membrane make different adsorption capacity for different oils.

4.      For the blend characterization, please refer to and cite related literature: Polymer Testing 2017, 59, 371-376; Advances in Polymer Technology 2018, 37 (7), 2609-2615.

5.      There are some grammar and format issues. The authors are suggested to polish the manuscript to decrease redundancy and correct the grammar mistakes.

Reviewer 2 Report

The manuscript of Yin Tang et al is devoted to the production of membranes made from fibers spun by electrospinning from mixed PS/PA6 systems. The theoretical part of the manuscript, in my opinion, can be expanded to understand the advantages of this particular system compared to others, for example, based on biopolymers. The service life of the described membrane is not clear, how the flows change with time, how the membrane will be cleaned from particles, etc. The novelty of the work should be more specifically formulated. Now the presented material looks more like an applied journal (The manuscript of Yin Tang et al is devoted to the production of membranes made from fibers spun by electrospinning from mixed PS/PA6 systems. The theoretical part of the manuscript, in my opinion, can be expanded to understand the advantages of this particular system compared to others, for example, based on biopolymers. The service life of the described membrane is not clear, how the flows change with time, how the membrane will be cleaned from particles, etc. The novelty of the work should be more specifically formulated. Now the presented material looks more like an applied journal (for instance Materials) than Polymers.   Line 14. "absorption" - maybe "adsorption"?! Line 121. Wrong dimension (g g-1). Line 170. "The emulsion-like immiscible precursor" is a bad start... Line 172. "The surface of PS/PA6 fibers was smooth" Lines 178-180. It is necessary to show the distribution of polymers, why are the authors sure that such a morphology has formed in the fibers? If the authors are talking about viscosity, then it makes sense to give its values. Figure 1. The diameter of the fibers is about 2-3 microns. Why are finer fibers not shown? Figure 3. It is necessary to describe in more detail the morphology for photographs a, d?! Figure 10. Need to add "angle".

Reviewer 3 Report

The manuscript is well organized and well written. The Abstract is clear but the introduction needs to be improved especially the aim of the study, It needs more relevant references   . The figures are also clear .However, the paper could be accepted after the minor revision.

Round 2

Reviewer 2 Report

Line 135. (g/•g-1) - remove dimension Figure 3. It is necessary to describe the formations on the fiber surface. Line 265. It is necessary to round all values, for example, 20.34 MPa.
